# A service evaluation of clinicians' Signposting of asylum seekers and refugees (ASRs) attending an emergency department in South-West England

Daniel Dolan[1], Assaf Givati [ID][2]*

1 University Hospitals Bristol and Weston NHS Foundation Trust, Bristol, United Kingdom, 2 Department of Population Health Sciences, King's College London, London, United Kingdom

* assaf.givati@kcl.ac.uk

## Abstract

Recent changes to health entitlements for asylum seekers and refugees (ASRs) in the UK may exacerbate existing barriers of access to healthcare for ASR patients and lead to greater numbers seeking emergency care. In the city in South-West England where this Formative Service Evaluation took place, there has been an increase in the population of ASR patients living in temporary accommodation and their usage of local health services is expected to increase. This Formative Service Evaluation evaluated clinicians' assessment and signposting of ASRs attending an Emergency Department in South-west England, by assessing three identified components of the service. It informed interventions that were then developed to improve the service. A mixed-methods survey was used to evaluate the service, whereby clinicians manage and signpost care for ASRs attending the Emergency Department. Purposive sampling was used to recruit 50 participants, including 30 medical and 20 nursing staff. Quantitative survey data was analysed using descriptive statistical analysis and qualitative data was analysed using conventional content analysis. This evaluation points at some inconsistencies of knowledge of ASRs health entitlements within the service; suboptimal practice of understanding and acting on barriers to ASRs' access to emergency service, and limited knowledge of local services available for ASR patients. Understanding the factors and circumstances affecting challenges to service provision to ASRs is important in developing strategies to improve this service.

## Introduction

The annual number of asylum applications to the UK rose steadily throughout the 2010s, and rapidly from 2021 onwards, reaching 81,130 applications in 2022, the highest annual number since 2002 [1]. In response, the UK government has implemented policy reducing entitlements to accessing public services for asylum seekers and refugees (ASRs) [2]. As part of this response, healthcare has been restricted, with changes to entitlements based on immigration status and new charges implemented [3]. Despite most ASRs

**Data availability statement:** All relevant data are within the paper and its Supporting Information files.

**Funding:** The authors received no specific funding for this work.

**Competing interests:** The authors have declared that no competing interests exist.

having exemptions from healthcare charges, lack of understanding of entitlements by both the ASR community as well as by the clinicians providing care, and difficulty proving exemption from fees, has led to inappropriate charging and denial of care [4].

Kang et al. [5] review identified a host of pre-existing factors and circumstances affecting the provision of health care to the ASR community, including (a) health service factors such as lack of knowledge about how the National Health Service (NHS) works and how to navigate and utilise its services, and confusion regarding entitlement to care for both clinicians and patients [6,7]; (b) language and communication factors such as availability of translation services and multilingual patient information or correspondence [8] as well as lack of time in primary care appointments to allow for communication needs [9] alongside a reduction in face-to-face appointments post-pandemic [10]; (c) cultural factors including lack of patient trust in healthcare systems due to fear of stigma, prejudice and lack of clinician understanding of their backgrounds and needs [11]; and (d) socio-economic factors including impermanence of location due to changes in dispersal accommodation preventing primary care registration and conditions of housing [12], costs of transport, cost of prescriptions or treatment [5] and social isolation [13].

These changes have the potential to compound existing barriers to access of healthcare for ASRs, and lead to inappropriate usage of services. As urgent care remains unrestricted, already overwhelmed Emergency Departments are likely to receive greater exposure to ASRs, and more regular attendances at urgent care facilities for lower acuity conditions [14]. The nature of ASRs' exposure to urgent care points at the importance of emergency clinician education regarding the health needs of the ASR community, so that they obtain appropriate guidance and health care. Clinicians' education about ASR health needs has been shown to improve the quality of care this community receives, and it has the potential to reduce unnecessary future presentations [15,16]. However, at present, there is no local or national guidance on the assessment or signposting of ASRs from urgent care or education for emergency clinicians.

### Local context and aim of service evaluation

Locally, the city where this Service Evaluation took place has seen increases in the numbers of ASRs, since 2022 hundreds of migrants have been placed in temporary accommodation, such as hotels, but with many more living in other forms of accommodation within the city [17]. It is therefore expected that there will be greater numbers of ASR patients attending health services in the area, including the Emergency Department in this Service Evaluation.

This formative service evaluation was designed to evaluate clinicians' signposting of ASRs attending the Department by assessing three components of the service: (a) clinicians' knowledge of ASR health entitlements; (b) Clinicians' awareness and practice relating to existing barriers to access of healthcare and (c) clinicians' awareness of local services for ASR patients. This data was then used to inform future evidence-based interventions that aim to improve the care and signposting of ASR patients from urgent care to other healthcare providers and third sector institutions in the community.

## Materials and methods

### Service evaluation design

This evaluation followed a formative design. Formative evaluation is set to assesses the development of a health programme while it is being developed and offer feedback to inform changes before the final product is completed [18]. The evaluation was informed by Stetler et al. [19] service evaluation model, which introduces four categories of "influences" on practice against which the Service Evaluation is performed:

Actual degree of less-than-best practice,

Determinants of current practice,

Potential barriers and facilitators to practice change and to implementation of the adoption strategy,

Strategy feasibility, including perceived utility of the project.

A mixed-methods survey (S1 Appendix; S2 Appendix) was used to gather clinicians' knowledge over four categories of influence: health entitlements of ASR patients (Section A); clinicians' practice in relation to barriers to ASR access to services (Section B); clinicians' knowledge of local healthcare and social support services for the ASR community, including the first assessment General Practice (GP) (in the UK, a GP provides primary healthcare to people of all ages and is often the first point of contact for patients) service in the city as well as any other third sector services available (Section C); and an invitation for clinicians to provide suggestions for interventions to improve the service (Section D). The data collection strategy is illustrated in Table 1:

Questionnaires were disseminated electronically, to achieve an optimal response rate [20]. Questionnaires were completed anonymously but included indication of role and seniority for the purpose of analysis. In the absence of a validated questionnaire, the questions were informed by tools used in similar improvement projects, which are the *Doctors of the World "Safe Surgeries" toolkit* [21] which was developed to support similar quality projects in primary care, and the British Medical Association's *Refugee and asylum seeker patient health toolkit* [22] developed as a resource for improving clinician knowledge of health entitlements of ASR patient and the challenges and barriers that they face.

### Data collection and sampling

Data collection took place between 01/12/2022 and 31/01/2023. The population recruitment included doctors and registered nursing staff in the ED. These are the clinicians that have the responsibility to assess, signpost and/or discharge patients. Purposive sampling was used. Clinicians in the Department were invited to take part in the evaluation through Departmental social media platforms, posters in both clinical and non-clinical areas, and announcements at meetings within the Department. The sample included 50 participants, 20 registered nurses and 30 medical staff, stratified into three seniority groups: group A, 15 participants with medical grades FY2-ST3 and nursing bands 4–5; group B, 11 participants, with medical grades ST4–6 and nursing band 6; and group C, 18 participants, consultants and nursing bands 7–8. FY2-ST3 refers to medical grades in the UK's training pathway, where FY2 is the final year of the Foundation Programme after medical school, and ST3 marks the beginning of higher specialty training after core training is completed. Nursing professional banding refers to the NHS (National Health Service) Agenda for Change pay structure, which categorizes jobs based on skills, experience, and responsibilities to determine salary and career progression.

### Data analysis

The questionaries generated nominal, ordinal and qualitative data (S1 Appendix). Quantitative data was analysed using descriptive statistical analysis and qualitative data collected from each section of the questionnaire was analysed using

**Table 1. Conceptualising influences central to data collection, adapted from Elwy et al.**

| Influences | Service Evaluation Methodology and Data Collection Strategy |
|---|---|
| Actual degree of less-than-best practice<br><br>Determinants of current practice | (a) Assessment of the current knowledge of clinicians regarding entitlements by using closed questions with reference to the current legal framework for ASR accessing emergency, primary and secondary care (S1 Appendix; S2 Appendix)<br>(b) Current clinician practice in relation to barriers of access for ASRs were assessed using questionnaires with open and closed questions based upon findings from literature review (S1 Appendix; S2 Appendix).<br>(c) Awareness of services in the locality for the community were evaluated using open and closed questions. |
| Potential barriers and facilitators to practice change and to implementation of the adoption strategy | Understanding the barriers to implementation and adoption of any interventions developed by this service evaluation is key to developing and improving the service.<br>Using open questions within each section to gain an understanding of the clinician perspective and providing opportunities for clinicians to suggest possible interventions. |
| Strategy feasibility, including perceived utility of the project | Stakeholder involvement was also seen to be integral to ensuring feasibility and utility, including representatives from the General Practice service dedicated to ASRs locally, senior medical and nursing staff – therefore including experts in providing care for the ASR community as well as experts in emergency care provision. |

*conventional content analysis* [23]. This basic technique was chosen due to its usefulness when analysing limited and discrete qualitative data and was undertaken using Microsoft Excel as a five-step process:

1. Familiarisation with data: the text obtained from the questionnaire was read in its entirety twice to achieve immersion and obtain a sense of the whole data.

2. Initial coding: each answer was read word by word, while highlighting key words that appear to capture core thoughts and experiences. At this point, initial impressions were noted.

3. Review of coding: emerging codes were labelled, reflective of more than one key thought/idea expressed within the answers.

4. Categorisation: the codes were then sorted into categories, based on how different codes are related and interlinked.

5. These emergent categories were then used to organize and group codes into meaningful clusters.

## Ethics statement

The Service Evaluation framing of this project was considered with advice from the host university, King's College London Research Ethics Committee (REC), and from the local Trust, UK Clinical Effectiveness and Audit manager. Ethical principles were followed, with consideration for voluntary participation, informed consent, potential for harm, anonymity and confidentiality (see Table 2).

According to the host university King's College London Research Ethics Committee (REC) guidance, "*Service Evaluation is undertaken to benefit those who use a particular service and is designed and conducted solely to define or judge current service. Your participants will normally be those who use the service or deliver it. It involves an intervention where there is no change to the standard service being delivered (e.g., no randomisation of service users into different groups). This does not require ethical review process*". The clinical effectiveness project application was submitted and approved by the Quality Governance Team of the NHS trust in which this project was undertaken on 16/12/2022 (project application number CE50833). Table 2 is a summary of ethical measured taken:

## Results

### Section A: Awareness of health entitlements of ASR patients

Forty percent (20/50) of participants reported knowledge of the UK government categorisation of immigrations statuses, including 25% (5/20) of nurses and 50% (15/30) of doctors. Sixty-two percent (31/50) stated they could define at least one

**Table 2. Ethical measures and justifications.**

| Evaluation Phase | Areas of Potential Ethical Concern | Measures taken |
|---|---|---|
| Research question/ problem definition | Service evaluation – ensuring benefit to the service | Utilising questions within the questionaries to ensure feedback regarding effectiveness of interventions and the need for interventions' Stakeholder involvement. |
| Research Design | Voluntary participation and consent | Voluntary nature of participation was clearly stated in all advertising and communications regarding the evaluation. Participant Information Sheets were included in the questionarie(S3 Appendix). The completion of the questionnaire was accepted as evidence of consent to participate in this service evaluation. The contact details of the evaluation lead were provided for any further questions about the evaluation. |
| | Confidentiality | No identifiable data was recorded for participants. The evaluation lead did not disclose any identifiable information of any candidate for any reason. |
| | Data security | Data was kept on a secure drive with encryption and kept in a locked drawer of a single office. |
| | Risk of harm to participants | Risk of harm to participants was minimal. By anonymising all the data as well as securing it, there was a minimal risk of this data becoming available to colleagues. |
| The researcher | Professional role within the Department | The lead of the SE had previously worked within the Department as a doctor but has left Department to work in another hospital and NHS trust. The role of the of the SE was junior and therefore unlikely to be felt as coercive. |
| | Knowledge and skills | The lead researcher ensured adequate research of the topic including undertaking a systematic review prior to planning the present evaluation. Evaluation and research techniques employed were informed by academic and research literature, and reviewed and discussed with the academic supervisor . |
| Outcome | Participant exclusion | Those members of staff not deemed registered nurses or doctors may feel excluded. It was therefore made clear that participants have been chosen for their specific role they play in relation to ASRs. |

of the UK immigration statuses with asylum seeker being the most commonly identified including 63% of doctors (19/30), 60% of nurses (12/20). Fifty-eight percent (24/50) of participants, including 60% (12/20) of nurses and 50% (15/30) of doctors were aware that ASRs immigration status can determine the nature of health care they are provided with. Twenty two percent of participants (11/50), including 23% (7/30) of doctors and 20% (4/20) of nurses, were able to correctly indicate which groups can access free primary care services. Seventy-four percent (37/50) of participants, 60% of nurses (12/20) and 86% (26/30) of doctors, were able to correctly identify which groups are entitled to free emergency care services. Thirty-two percent (16/50) of participants, 25% (5/20) of nurses and 36% (11/30) of doctors, were able to correctly identify which groups are entitled to free secondary care services. Finally, sixty-eight percent (34/50) of participants, 55% (11/20) of nurses and 76% (23/30) of doctors, were aware that patients do not need to provide photographic identification to attend an emergency Department. However, only sixteen percent (8/50) of participants, 10% (2/20) of nurses and 20% (6/30) of doctors, were aware that patients do not need to provide photographic identification to register at a General Practice.

## Section B: Clinicians' practice in relation to barriers of ASR access to services

Ninety-two percent (46/50) of clinicians, 93% (28/30) doctors and 90% (18/20) of nurses stated they never or rarely ask about immigration status. Forty-two percent (21/50) of participants, 53% (16/30) of doctors and 25% (5/20) of nurses felt they "definitely" or "probably" should not ask about immigration status routinely. Conversely, twenty-two percent (11/50) of participants, 20% (4/20) of nurses and 23% (7/30) of doctors, consider routine enquiry about immigration status. Moreover, twenty percent (10/50) of participants, 20% of nurses (4/20) and doctors (6/30), suggested they would rarely or never ask if a patient was registered at a General Practice. In contrast, 44% (22/50) of participants, 37% (11/30) doctors and 55% (11/20) of nurses, would often or always ask if a patient was registered at a General Practice. Overall, 62% (31/50) of clinicians felt they had, at least sometimes, assessed an ASR patient who attended or directed to urgent care inappropriately.

Concerning the role of translation as part of the medical encounter with ASRs, 53% (16/30) of doctors and 35% (7/20) of nurses always or often use the trust's translation service, whilst 46% (14/30) of doctors and 45% of nurses (9/20) often use family or friends, and 22% (11/50) of participants often use google translate. In addition, 33% (10/30) of doctors and 25% (5/20) of nurses felt that cultural differences always or often affect the care they give patients, with 56% (28/50) experiencing such cultural differences in practice "sometimes" and only 14% (7/50) "rarely or never" experiencing it. Finally, 20% (10/50) of all participants felt that immigration status often or always affects their prescribing/administering practices, for example giving TTAs vs FP10 or referral to General Practice. In the NHS, "TTAs" stand for "To Take Away/Out" which are a type of hospital discharge medication summary, distinct from the standard "FP10" prescription form which is used in primary care and from or the formal referral to General Practice. Each serve different purposes and have different legal/procedural requirements.

## Section C: Clinicians' knowledge of local services for the ASR community including the first assessment General Practice service in the city as well as any other third sector services available

Seventy-six percent of participants (33/50), 67% (20/30) of doctors and 90% (18/20) of nurses were not aware of any local services for ASR patients, higher in groups A (80%) and B (88%) and lowest in group C (61%). Of the participants who were aware of local services for ASR patients, sixteen percent (8/50) of participants, 23% (7/30) of doctors and 5% (1/20) of nurses, identified the local first-assessment General Practice, with no participants identifying other health or social support services.

## Section D: An invitation for clinicians to provide suggestions for interventions to improve the service

Ninety-four percent of clinicians (47/50), including 90% (27/30) of doctors and 100% (20/20) of nurses, felt that education about ASR health, entitlements and signposting would be valuable. Most indicated that teaching sessions as well as

resources and guidance for clinicians would be useful. Similarly, 96% (48/50) of all participants thought that resources within the Department detailing local services available to ASRs would be valuable. Most indicated that information packs for patients or online resources to aid signposting is needed.

### Findings obtained from open questions

Most of the open-ended answers were developed to provide an insight into clinician practice and signposting of ASR patients. Broad themes emerged relating to patient factors, clinician factors and system factors, summarised in Table 3. These findings provide local contextualisation to the barriers identified in the introduction, but also highlights more nuanced areas in practice that may be targeted:

### Summary of Key findings

These findings point at inconsistencies and gaps in knowledge of emergency clinicians regarding immigration statuses and health entitlements of ASRs. There was also a notable lack of knowledge of identification requirements to register at a General Practice service and a significant proportion believing that ID is required to seek emergency care. Seniority does not seem to impact knowledge consistently, although there is some indication that knowledge of entitlement to emergency care and ID requirements improves with grade, which may reflect clinical experience.

The evaluation shows variation and less-than-best practice in clinicians' awareness of barriers to healthcare access for ASR patients. Most of the study participants do not routinely enquire about immigration statuses, often out of fear of causing offence, but also due to time constraints around recording and interviewing patients. Clinicians suggested several system factors behind poor primary care registration rates for the ASRs, such as resource constraints, but also patient-factors including language skills and lack of knowledge of the health system. Generally, there is poor uptake of the translation service, however doctors reported using this more than nursing colleagues, possibly relating to the detailed history-taking required. Other methods such as seeking support from family members or friends to translate information, or the use of Google Translate, were also mentioned by the study participants. Resource constraints, particularly time availability, was the primary reason for choice. The findings also confirm that cultural barriers play significant role in the consultation of ASRs, and clinicians cited perceived over-reaction to advice, frustration with resource constraints and misaligned patient expectations.

Finally, more than two thirds of clinicians were not aware of any services available for ASRs in the local area. This shows a gap in knowledge of the services available to ASRs and a significant area for intervention if signposting was to improve. There is also evidence of an appetite for education amongst clinicians with regards to ASR health, as well as an awareness of the need for better guidance and resources for ASR patients to improve signposting.

## Discussion

It has been shown that the complexity of recent changes to ASR entitlements has led to misunderstandings about eligibility for accessing care by both ASRs and clinicians[7]. Healthcare workers often lack awareness of the definitions of immigration categories upon which entitlements are based [24]. There is evidence that General Practitioners have

Table 3. Qualitative themes and code categories.

| Patient factors | Clinician factors | Environment and system factors | Interventions |
|---|---|---|---|
| - Fear of patient offence<br>- Social circumstances<br>- Cultural and language barriers | - experience with translation services<br>- Clinician knowledge | - Change inertia<br>- Resource constraints<br>- Impact on ongoing care | - Formal teaching<br>- Education resources guidance<br>- Evaluation<br>- Clinical knowledge<br>- Patient resources |

refused registration to ASRs due to misunderstandings of the need for ID and address checks [25]. A member survey conducted by the British Medical Association showed that more than half of doctors regularly in contact with ASRs are uncertain about their entitlements [6]. There is also a lack of understanding of the NHS and how it operates within the ASR community [5].

A review of the literature provided a basis for assessing practice in relation to known barriers to access of health-care for ASR patients in this evaluation. Pollard et al. [8] showed that translation services are generally variable in quality and engagement. Healthcare professionals using family or friends to translate, which may be ineffective or inappropriate, has been identified as a barrier to access and provision of quality care [13]. The need for increased time for assessment by clinicians has also been an obstacle [9]. There are indications that fear of stigma or racial prej-udice prevents some from the ASR community from attending health and social care services [26]. Additionally, finan-cial barriers have been identified, particularly costs associated with transport and prescription costs [5] or being able to afford childcare in order to attend health appointments, as well as social factors including social isolation and lack of social support network [13]. Impermanence in location due to the government dispersal, moving ASRs to various temporary residences, prevents healthcare continuity and General Practice registration [12]. Emergency Departments have greater exposure to the ASR communities than other healthcare institutions, in part due to the aforementioned factors [14]. These factors could prevent ASR patients presenting to primary care when needed and lead to a greater reliance on urgent care. While there has been little research within emergency care settings, it has been shown that upskilling and educating clinicians about ASR health needs and barriers to access, can improve the quality of care as well as reduce unnecessary presentations [15,16].

### What this Evaluation adds to this service?

Evidence-based interventions have been developed using this local data with a view to improve the service being pro-vided. It is hoped that the interventions will improve clinician knowledge and provide local guidance and resources to inform decision making and signposting of ASR patients. When considering implementing healthcare interventions it is important that there is an adequate evaluation process. Therefore, an ongoing quality improvement project is being developed with the aim of rolling these interventions out whilst evaluating their impact and considering how they can be improved and adapted. It is recommended that further work to engage the ASR community directly, particularly those in temporary housing, to enhance understanding of the health system and entitlements including usage patterns and bar-riers to access locally. Table 4 provides a summary of interventions derived from the review in relation to each service component:

### Limitations

An apparent limitation of this evaluation was the lack of participation of the ASR community, the service users, and not being able to include their experiences of seeking care locally. This was not seen to be possible in this case due to the need to involve ASRs ethically, sensitively and effectively, which would require to undertake as separate *research* (rather than a *Service Evaluation*).

### Conclusions

This formative service evaluation assessed the signposting of ASRs by the Emergency Department where the evaluation took place in relation to (a) the knowledge of the clinicians regarding health entitlements; (b) the knowledge and practice of clinicians relating to known barriers to access of healthcare; and (c) clinician awareness of the local services. The eval-uation illustrates limited clinician knowledge of ASRs' health entitlements and of the local ASRs services. It identified areas for enhancing those parts of the service and it proposed interventions to remedy some of these concerns. Developed interventions include guidance for clinicians when assessing ASR patients, clinician education including formal teaching

**Table 4. Summary of Evidence Based Interventions Developed for Each Component of the Service.**

| Area of concern | Proposed interventions |
|---|---|
| 1. Immigration status and health entitlements | • Brief teaching sessions covering a) basic knowledge of immigration status, entitlements and barriers to access; b) introduction to guidance and services for signposting. Ideally, each 6 months to cover doctors' rotation.<br>• Educational resources for clinician on intranet and dedicated app. provided by British Medical Association, and the National Health Service, including quire reference guide to ASRs health entitlements. |
| 2. Barriers to access of healthcare | • Brief teaching sessions as above.<br>• "One-Stop" guidance document for clinician, a quick reference guide for entitlements as well as well as signposting directions. |
| 3. Signposting and patient education | • Patient information pack that can be translated online, containing details of local services.<br>• Referral information for clinicians: referral forms for new patients to first-assessment General Practice service and email address for already registered patients to improve information sharing. |

and resources and patient information packs. Future recommendations include an evaluation of service with the ASR community to understand their experiences and views of the service provided.

## Supporting information

**S1 Appendix. Complete data from questionnaire analysed for the present service evaluation including raw qualitative data.**
(PDF)

**S2 Appendix. Structured Questionnaire used for data collection.**
(DOCX)

**S3 Appendix. Participant information sheet provided to participants before completing the questionnaire.**
(DOCX)

## Acknowledgments

By "signposting" we refer to directing/guiding asylum seekers and refugees in relation to their health care entitlements and related services.

## Author contributions

**Conceptualization:** Daniel Dolan, Assaf Givati.

**Data curation:** Daniel Dolan.

**Formal analysis:** Daniel Dolan.

**Investigation:** Daniel Dolan.

**Methodology:** Daniel Dolan, Assaf Givati.

**Project administration:** Daniel Dolan.

**Supervision:** Assaf Givati.

**Writing – original draft:** Daniel Dolan.

**Writing – review & editing:** Assaf Givati.

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
