## [Decision Letter · Decision Letter 0]

24 Oct 2025

PGPH-D-25-02560

A Service Evaluation of Clinicians’ Signposting* of Asylum Seekers and Refugees (ASRs) attending an Emergency Department in South-West England

Dear Dr. Givati,

Thank you for submitting your manuscript to PLOS Global Public Health. After careful consideration, we feel that it has merit but does not fully meet PLOS Global Public Health’s publication criteria as it currently stands. Therefore, we invite you to submit a revised version of the manuscript that addresses the points raised during the review process.

Kindly attend to the comments that the reviewers have provided paying attention to spelling and ensuring that the paper is copyedited before resubmission.

We look forward to receiving your revised manuscript.

Kind regards,

Ferdinand C Mukumbang, PhD

Academic Editor

Journal Requirements:

1. Please send a completed 'Competing Interests' statement, including any COIs declared by your co-authors. If you have no competing interests to declare, please state "The authors have declared that no competing interests exist". Otherwise please declare all competing interests beginning with the statement "I have read the journal's policy and the authors of this manuscript have the following competing interests:"

3, We have noticed that you have uploaded Supporting Information files, but you have not included a list of legends. Please add a full list of legends for your Supporting Information files after the references list.

4. In the online submission form, you indicated that The data supporting a study's findings are not publicly accessible but can be obtained by other researchers if they make a reasonable request to the authors.

3. Uploaded as supplementary information.

Reviewers' comments:

Reviewer's Responses to Questions

**Comments to the Author**

1. Does this manuscript meet PLOS Global Public Health’s publication criteria?

Reviewer #1: Partly

Reviewer #2: Yes

2. Has the statistical analysis been performed appropriately and rigorously?

Reviewer #1: Yes

Reviewer #2: Yes

3. Have the authors made all data underlying the findings in their manuscript fully available (please refer to the Data Availability Statement at the start of the manuscript PDF file)?

Reviewer #1: No

Reviewer #2: No

4. Is the manuscript presented in an intelligible fashion and written in standard English?

Reviewer #1: Yes

Reviewer #2: Yes

Reviewer #1: I want to thank the authors for taking their interest on this important topic. In the last decade, the increasing number of ASR makes this topic of health service provision a very relevant one. The manuscript can be made stronger with a few adjustments.

1. The manuscript is full of abbreviations, which makes the reading flow difficult. e.g UK, GP, SE, BMA, FY2-ST3, TTA, FP10, ED, NHS etc. Please have a look though out the manuscript and adjust accordingly.

2. The manuscript has a number of spelling mistakes, a copy editing process might be useful. Please check a few examples here with highlighted from the manuscript: Page 9 Evaluation phase (research design) please have a look at the second last sentence in this box, Page 11 line 213, page 10 in the table (The researcher) please check the second sentence in the last column. In the table (Outcome), please check the last sentence, page 15, please check line 296, page 14, please check line 289.

3. Reference documents such as Safe Surgery tool kit and refugee and asylum seeker patient health tool kit could be provided as part of the annex or part of the supporting information.

4. Under data analysis, the process and steps taken are clear for both quantitative and qualitative. However for the qualitative, it is important to highlight the five steps and the role of the second person for triangulation of information.

5. Page 8, under ethical statement, do we have a similar response from Bristol Hospital as the Kings College London?

6. Page 8 line 179, reads incomplete on its own. Considering this is also in the table, may useful to rewrite or remove it completely.

7. Page 9, please check if possible to increase the font size in the table.

8. Page 10, you are repeating what is already stated on page 8, under ethics statement.

9. Page 10, the statement in the table under knowledge and skills, It is important to focus on the skills of the research team, instead of writing or mentioning the supervisor, it is important to mention the skills of the supervisor (research team).

Reviewer #2: An applaud to the authors for this interesting manuscript that focused on "A Service Evaluation of Clinicians’ Signposting* of Asylum Seekers and Refugees (ASRs) attending an Emergency Department in South-West England".

INTRODUCTION

-Line 64: Consider changing “have” to “has”

-Line 72 and throughout the manuscript: consider correcting “et al” to “et al.”

-Line 105: For consistency, please check how Department is written in the lines above

-Line 108: Consider changing “Clinician” to “Clinicians”

-Line 151: Consider changing “recruited” to “recruitment”

-Line 189: Table 2, under the section Measures taken for Research and design, consider changes “The contact details of the evaluation lead were be provided for any further questions about the evaluation” to “The contact details of the evaluation lead were provided for any further questions about the evaluation”

RESULTS

General comment

Given the small number of study participants relative to the countries population of medical staff, I would recommend that the authors should add the number of staff for each result i.e. x/x through the results.

-Line 206: Consider moving the heading one line above

- Line 213: delete duplicate word “that”

-Line 241: Consider changing “using” to “use”

-Line 258: Please check, either delete "and" or add what seems to be missing

-Line 277: Consider moving this to two lines above

-Line 289: Consider changing “acquire” to “enquire”

-Line 325: remove the words “seen to be”

-Line 354: Capitalize the word “Summary”

Line 407-409: Inconsistency with word capitalization. Authors should stick to one style of writing reference titles.

-Line 468: For consistency with other references, authors should consider capitalizing the letter after the colon (:).

**Do you want your identity to be public for this peer review?** For information about this choice, including consent withdrawal, please see our Privacy Policy

Reviewer #1: **Yes:** Norman Sitali RN/MPH/MIH

Reviewer #2: **Yes:** Naomi Chi Ndum

---

## [Editor Report · Decision Letter 1]

14 Dec 2025

A Service Evaluation of Clinicians’ Signposting* of Asylum Seekers and Refugees (ASRs) attending an Emergency Department in South-West England

PGPH-D-25-02560R1

Dear Assaf Givati

We are pleased to inform you that your manuscript 'A Service Evaluation of Clinicians’ Signposting* of Asylum Seekers and Refugees (ASRs) attending an Emergency Department in South-West England' has been provisionally accepted for publication in PLOS Global Public Health.

Best regards,

Ferdinand C Mukumbang, PhD

Academic Editor

Consider reading the manuscript to address all grammatic concerns. For instance, in abstract the authors write "data was". Data is the plural form of datum, thus should be "data were".